

# A hybrid feature selection algorithm and its application in bioinformatics

Yangyang Wang[1], Xiaoguang Gao[1], Xinxin Ru[1], Pengzhan Sun[1] and Jihan Wang[2]

[1] School of Electronics and Information, Northwestern Polytechnical University, Xi'an, Shaanxi, China
[2] Institute of Medical Research, Northwestern Polytechnical University, Xi'an, Shaanxi, China

## ABSTRACT

Feature selection is an independent technology for high-dimensional datasets that has been widely applied in a variety of fields. With the vast expansion of information, such as bioinformatics data, there has been an urgent need to investigate more effective and accurate methods involving feature selection in recent decades. Here, we proposed the hybrid MMPSO method, by combining the feature ranking method and the heuristic search method, to obtain an optimal subset that can be used for higher classification accuracy. In this study, ten datasets obtained from the UCI Machine Learning Repository were analyzed to demonstrate the superiority of our method. The MMPSO algorithm outperformed other algorithms in terms of classification accuracy while utilizing the same number of features. Then we applied the method to a biological dataset containing gene expression information about liver hepatocellular carcinoma (LIHC) samples obtained from The Cancer Genome Atlas (TCGA) and Genotype-Tissue Expression (GTEx). On the basis of the MMPSO algorithm, we identified a 18-gene signature that performed well in distinguishing normal samples from tumours. Nine of the 18 differentially expressed genes were significantly up-regulated in LIHC tumour samples, and the area under curves (AUC) of the combination seven genes (ADRA2B, ERAP2, NPC1L1, PLVAP, POMC, PYROXD2, TRIM29) in classifying tumours with normal samples was greater than 0.99. Six genes (ADRA2B, PYROXD2, CACHD1, FKBP1B, PRKD1 and RPL7AP6) were significantly correlated with survival time. The MMPSO algorithm can be used to effectively extract features from a high-dimensional dataset, which will provide new clues for identifying biomarkers or therapeutic targets from biological data and more perspectives in tumor research.

# INTRODUCTION

The dimensionality of data has increased greatly due to the rapid growth in big data (*Li, Wu & Li, 2020*; *Wainwright, 2019*). This condition has also accelerated the development of high dimensional data processing technology (*Li et al., 2016*; *Saeys, Inza & Larrañaga, 2007*). One of the main issues in data mining, pattern recognition, and machine learning is feature selection for high dimensional data (*Chen et al., 2020*; *Larranaga et al., 2006*). Feature selection is the process of selecting the feature subset that best captures the characteristics of the original dataset and alters the feature expression of the original dataset as little as possible. It can be utilized as an important dimensionality reduction

Corresponding authors
Xiaoguang Gao, xggao@nwpu.edu.cn
Jihan Wang, jihanwang@nwpu.edu.cn

technique to minimize computing complexity, lower the potential of overfitting as well as improve the prediction performance (*Tao et al., 2015*). Feature selection seldom modifies the original feature space, and the resultant feature subset has clearer physical implications that can be exploited for subsequent classification or inference (*Villa et al., 2021*). The search for the optimal subset of features is typically computationally expensive and has been demonstrated to be nondeterministic polynomial-hard (NP-hard) (*Faris et al., 2018*; *Wang, Wang & Chang, 2016*). Traditionally, feature selection algorithms are classified into three categories: filter, wrapper, and embedded methods and these methods can also be divided into two main categories: feature ranking and feature subset selection (*Van Hulse, Khoshgoftaar & Napolitano, 2011*). In the past few years, feature selection based on high-dimensional datasets has attracted more attention. Because of their simplicity and efficiency, ranking-based approaches such as ReliefF (*Robnik-Šikonja & Kononenko, 2003*), minimum-redundancy maximum-relevancy (mRMR) (*Peng, Long & Ding, 2005*), Fisher (*Gu, Li & Han, 2012*), CFS (*Zeng & Cheung, 2010*), and others are widely utilized in a variety of applications. Different from the feature ranking selection, which screens out the top $K$ highest-scoring features, feature subset selection selects the subset of features that perform well together. Some heuristic search strategies (*Rasheed, 2021*) have been proposed to obtain the global optimal feature subset, such as the genetic algorithm (GA) (*Holland, 1975*; *Stefano, Fontanella & Freca, 2017*), particle swarm optimization (PSO) (*Chuang et al., 2008*; *Eberhart & Kennedy, 1995*; *Wang et al., 2007*), and ant colony optimization (ACC) (*Li, Wang & Song, 2008a*). It is worth mentioning that, some methods based on neural networks which supports higher-dimensional inputs can also be used for feature selection (*Liu, Liu & Zhang, 2022*). Feature selection has been widely utilized in bioinformatics to remove irrelevant features in high-throughput data as an effective method for preventing the "curse of dimensionality" (*Li et al., 2008b*). It is appropriate to filter out biomarkers in the medical field, which can not only help explore disease pathophysiology at the molecular level but also has advantages in accurate diagnosis. In general, the number of features in a bioinformatics dataset tends to be very large. It is critical to identify highly discriminating biomarkers to improve disease diagnosis and prediction accuracy (*Ma et al., 2020*). Therefore, there is no doubt that obtaining relevant biomarkers from high-throughput data is of great significance (*Han, Huang & Zhou, 2021*). Furthermore, we realized that there is considerable space for improvement in the feature selection process by combining feature ranking with feature subset searching. There are several methods for measuring the specific value of relevance, including the Pearson correlation coefficient (*Obayashi & Kinoshita, 2009*), mutual information (MI), and maximum information coefficient (MIC) (*Reshef et al., 2011*), and MIC can substitute MI to obtain better mutual information measurement results in some situations, particularly for continuous data. Furthermore, feature ranking does not provide a "golden standard" for obtaining the best feature subset but only the ranking result. Therefore, combining the two methods is a more promising way (*Shreem et al., 2013*; *Stefano, Fontanella & Freca, 2017*).

In this study, we focus on developing a hybrid efficient approach for obtaining the optimum features by combining the feature ranking method and the heuristic search method. Specifically, ten datasets were employed to validate our hypothesis first.

Furthermore, one dataset derived from high-throughput sequencing was used to assess the effect of the approach at the genetic level. The discussion and conclusion are presented in the last section.

## MATERIALS & METHODS

### The mRMR algorithm

The mRMR (*Peng, Long & Ding, 2005*) algorithm, which uses mutual information to assess the relevance between features, has been used in bioinformatics (*Ding & Peng, 2005*; *Li et al., 2012*; *Mundra & Rajapakse, 2010*). Mutual information is widely used to analyze the correlation between two variables, and it can be expressed as Eq. (1).

$$I(X,Y) = \sum_{x \in X} \sum_{y \in Y} p(x,y) \log \frac{p(x,y)}{p(x)p(y)}. \tag{1}$$

In the Eq. (1), P represents the probability and $X$, $Y$ represent the feature vector or class vector. The relevance $V$ and redundancy $W$ of the mRMR can be expressed using Eqs. (2) and (3).

$$V = \frac{1}{|S|} \sum_{x_i \in S} I(y; x_i). \tag{2}$$

$$W = \frac{1}{|S|^2} \sum_{x_i, x_j \in S} I(x_i; x_j). \tag{3}$$

In the Eqs. (2) and (3), $y$ is the target variable, $S$ is candidate feature set and $x_i$, $x_j$ are arbitrary variables of $S$. To calculate the final score of relevance, the *MIQ* can be used, as shown in Eq. (4).

$$MIQ : argmax\left(\frac{V}{W}\right). \tag{4}$$

### Maximal information coefficient

As a measure of dependence for two-variable relationships in a large dataset, MIC has been widely used in various fields, including global health, gene expression, human gut microbiota and identify novel relationships due to its ability to capture a wide range of functional and non-functional associations. The definition of MIC is:

$$MIC(x,y|D) = max_{i*j<B(n)} \left\{ \frac{I^*(x,y,D,i,j)}{\log \min(i,j)} \right\}. \tag{5}$$

In the Eq. (5), $x$ and $y$ are the *pairs(x, y)* of the dataset and $I^*(x, y, D, i, j)$ denotes the maximum mutual information of $D|_G$ with the *i-by-j* grid, where the default $B(n) = n^{0.6}$.

## Particle swarm optimization algorithm

The particle swarm optimization (PSO) algorithm is a heuristic search algorithm that originated from studies of bird predation behavior (*Eberhart & Kennedy, 1995*). The first step of PSO is to initialize a group of particles. It then iterates until it finds the best solution. The particles update themselves in each iteration by tracking two extreme values. The first is the particle's determination of the individual extreme value *Pbest*. The other is called *Gbest*, which is determined by the entire particle swarm. After determining the *Gbest* and *Pbest* values, the particle updates its speed and position based on Eqs. (6) and (7).

$$V_{k+1} = \omega V_k + c_1 r_1 (Pbest - X_k) + c_2 r_2 (Gbest - X_k). \tag{6}$$

$$X_{k+1} = X_k + V_k. \tag{7}$$

In the above equations, $k$ represents the number of iterations; $V_k$ and $X_k$ represent the particle's current velocity and position, respectively; $r_1$, $r_2$ are random values between [0, 1]; $c_1$, $c_2$ are the learning factors; $\omega$ is the inertia weight, which is used to control the influence of the last iteration's speed on the current speed. A smaller and larger $\omega$ can strengthen the PSO algorithm's local or global search ability, respectively.

## The hybrid algorithm for feature selection

In this study, we proposed the MMPSO hybrid method. First, the dataset needed to be preprocessed. On the one hand, the aim of preprocessing is to remove some features that contain a large quantity of noisy data, such as features that contain many zeros. On the other hand, if the proportion of samples is clearly unbalanced, it is necessary to balance the samples. Here, we employed random oversampling technology to address this issue. A random over-sampler randomly copies and repeats the minority class samples, eventually resulting in the minority and majority classes having the same number. The next step was to rank the features. MIC was used to measure the correlation of two features, resulting in a more accurate ranking result of features based on the mRMR framework (*Cao et al., 2021*). Considering the numerous features and the complexity of MIC, we used the multithreading method in paper (*Tang et al., 2014*) to speed up the calculation. After performing the mRMR based on MIC method, we obtained the ranking features and use the top $K$ features as the input of the next step to reduce the computational load for the PSO. The $K$ features were used in the third step to initialize the particle swarm and calculate the fitness of each particle. For the wrapper feature selection algorithm, only the classification accuracy is used as the fitness function to guide the feature selection process, which will lead to a larger scale of the selected feature subset (*Liu et al., 2011*). Therefore, some studies combine the classification accuracy and the number of selected feature subsets to form a fitness function (*Xue, Zhang & Browne, 2012*). Here, the fitness we defined is shown in Eq. (8). $V_{error}$ was the error, which is measured by a classification method of k-nearest neighbor (KNN) (*Chen et al., 2021a*). $N_{selected}$ and $N_{all}$ were the numbers of selected features and the entire features, respectively. $\alpha$ and $\beta$ were parameters whose sum is 1. The larger $\alpha$ is, the more features will be chosen; otherwise, fewer features will be

chosen. When the specified number of iterations is reached, the PSO program terminates, and the final selected features will be available.

$$cost = \alpha V_{error} + \beta \frac{N_{selected}}{N_{all}}. \tag{8}$$

### The validation method

Here, we respectively compared the classification accuracy of the MMPSO method with the results of other algorithms, including mRMR (*Peng, Long & Ding, 2005*), ILFS (*Roffo et al., 2017*), ReliefF (*Liu & Motoda, 2007*), Mutinffs (*Zaffalon & Hutter, 2002*), FSV (*Bradley & Mangasarian, 1999*), Fisher (*Gu, Li & Han, 2012*), CFS (*Zeng & Cheung, 2010*), UFSOL (*Guo et al., 2017*), to demonstrate that our method has better classification accuracy. LIBSVM (*Chang & Lin, 2011*) is an integrated library, which supports multi-class classification. Here, we performed classification using LIBSVM to test the accuracy with k-fold cross validation.

### Summary of datasets

A total of eleven datasets were used in this study; the basic information about the datasets was shown in Table 1. The UCI Machine Learning Repository is a collection of databases, domain theories, and data generators that are used by the machine learning community for the empirical analysis of machine learning algorithms (*Dua & Graff, 2017*). In the beginning, ten datasets that have different numbers of features, instances and classes were downloaded from the UCI website, and they were used to evaluate the performance of our proposed method. Furthermore, we conducted a more thorough analysis to demonstrate the biological application of our method. Tremendous amount of RNA expression data has been produced by large consortium projects such as TCGA and GTEx, creating new opportunities for data mining and deeper understanding of gene functions (*Tang et al., 2017*). Thus, the final liver hepatocellular carcinoma (LIHC) dataset was obtained from UCSC Xean, which contains large-scale standardized public, multiomic and clinical/phenotype information (*Goldman et al., 2020*). The LIHC dataset used in the current study contains RNA expression data of over 60,000 genes in 531 biosamples (371 tumor samples and 160 normal samples, and the latter further containing 50 normal samples from the TCGA-LIHC cohort and 110 normal tissues from GTEx), and it is available at https://toil-xena-hub.s3.us-east-1.amazonaws.com/download/TCGA-GTEx-TARGET-gene-exp-counts.deseq2-normalized.log2.gz.

## RESULTS

In this section, we focused on testing the accuracy of our proposed MMPSO method and compared it with other methods of mRMR, ILFS, ReliefF, Mutinffs, FSV, Fisher, CFS and UFSOL. All experiments in this study were carried out on a Windows 10 system with an Intel(R) Xeon(R) CPU E5-2420, 1.9 Ghz processor with 16 GB RAM. Our proposed algorithm was implemented in MATLAB 2020b, and the PSO parameters were as follows: population size: 100; number of iterations: 50; $c_1$: 2; $c_2$: 2; $\omega$: 0.9; $\alpha$: 0.95; $\beta$: 0.05.

**Table 1  Basic information of the datasets in this study.**

| Datasets | | Instances | Features | Classes |
|---|---|---|---|---|
| Datasets obtained from UCI | gina | 3468 | 970 | 2 |
| | gisette | 6000 | 5000 | 2 |
| | hillvalley | 1212 | 100 | 2 |
| | isolet | 7797 | 617 | 26 |
| | madelon | 2000 | 500 | 2 |
| | musk | 6598 | 166 | 2 |
| | scene | 2407 | 294 | 2 |
| | splice | 3190 | 60 | 3 |
| | usps | 9298 | 256 | 10 |
| | waveform | 5000 | 21 | 3 |
| Biological dataset | LIHC | 531 | 60498 | 2 |

**Notes.**

The ten datasets obtained from UCI were analyzed to demonstrate the superiority of MMPSO; the biological dataset was used as an application of the proposed MMPSO method.

## Results of the experiment based on ten datasets

Figure 1 and Table 2 summarized the classification accuracy on the basis of the MMPSO and the compared methods. Here, we defined the threshold $K$ was 100. When the number of original features was greater than 100, the top 100 features from the ranking result were selected as the PSO input for the MMPSO method; otherwise, all features were selected as the input. In Fig. 1, we obtained the conclusion that our method was superior to the other methods in terms of classification accuracy based on six datasets including gisette, hillvalley, isolet, madelon, scene and usps. In the other four datasets, the MMPSO method achieved the accuracies of top three rankings. Therefore, the MMPSO algorithm was outperformed other methods with respect to accuracy of classification by utilizing the same number of features.

## Results of the experiment based on the biological dataset

After analyzing the first ten datasets, we employed the MMPSO method on the LIHC dataset to identify features (gene biomarkers) that can be used to distinguish the tumor group from the normal group with high accuracy. Different from the previous datasets, LIHC is an unbalanced dataset containing 371 tumor samples and 160 normal samples. Therefore, the preprocessing was needed and the number of genes was reduced from over 60,000 to 15,185 after preprocessing. When the genes were ranked by the mRMR based on MIC method, the top 100 genes were selected and input into PSO to identify the best gene signatures. Finally, we obtained a signature of 18 genes through the MMPSO algorithm, that had better classification compared to other methods. The 18 genes were ACTN1, CACHD1, ERAP2, FAM171A1, FKBP1B, HIST1H2BC, PLVAP, PRKD1, RPL7AP6, ADRA2B, DMKN, FNDC4, NPC1L1, POMC, PYROXD2, RBP1, TRIM29, and ZBED9, with the relative expression of the first nine genes significantly increasing in tumors and the last nine genes decreasing ($P < 0.01$ in the Wilcoxon rank sum test with continuity correction, Fig. 2). Principal component analysis (PCA) was then performed using the 'FactoMineR' and

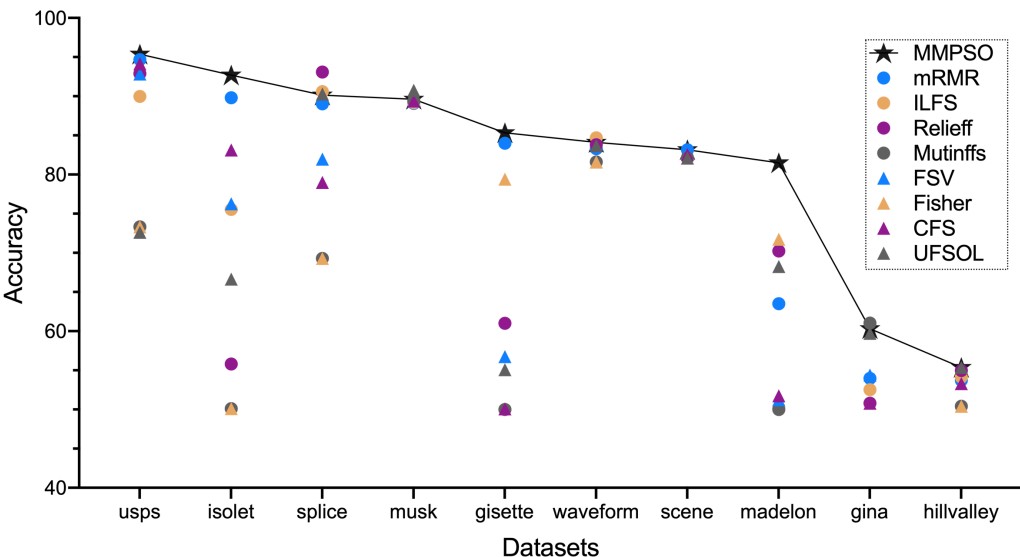

**Figure 1** The accuracy of nine algorithms based on ten datasets.

**Table 2** Accuracy of algorithms based on the ten datasets.

| Datasets | MMPSO | mRMR | ILFS | ReliefF | Mutinffs | FSV | Fisher | CFS | UFSOL |
|---|---|---|---|---|---|---|---|---|---|
| gina | 60.32 | 53.97 | 52.53 | 50.79 | **61.04** | 54.40 | 50.79 | 50.79 | 59.74 |
| gisette | **85.33** | 84.00 | 50.00 | 61.00 | 50.00 | 56.75 | 79.42 | 50.08 | 55.08 |
| hillvalley | **55.37** | 53.72 | 54.13 | 54.96 | 50.41 | 50.41 | 50.41 | 53.31 | 55.37 |
| isolet | **92.69** | 89.80 | 75.56 | 55.81 | 50.10 | 76.27 | 50.10 | 83.13 | 66.65 |
| madelon | **81.50** | 63.50 | 50.25 | 70.25 | 50.00 | 51.25 | 71.75 | 51.75 | 68.25 |
| musk | 89.61 | 89.08 | 89.16 | 89.39 | 89.39 | 89.39 | 89.39 | 89.39 | **90.75** |
| scene | **83.16** | 83.16 | 82.12 | 82.12 | 82.12 | 82.12 | 82.12 | 82.54 | 82.12 |
| splice | 90.13 | 89.02 | 90.60 | **93.10** | 69.28 | 81.97 | 69.28 | 79.00 | 90.28 |
| usps | **95.37** | 94.67 | 89.99 | 92.90 | 73.32 | 92.85 | 73.32 | 94.14 | 72.62 |
| waveform | 84.10 | 83.30 | **84.70** | 83.80 | 81.60 | 81.60 | 81.60 | 83.70 | 83.70 |

'Factoextra' packages in R version 4.0.2 based on the expression profiles of the candidate 18 genes. As shown in Figs. 3A–3B, Dim 1 and Dim 2 were 15% and 11.9%, respectively. Figure 3C illustrated the heatmap of all samples based on the 18 gene expression profiling. The results revealed that the 18-gene signature obtained from MMPSO algorithm could effectively separate the 531 samples into two groups. Since logistic regression show that seven biomarkers of ADRA2B, ERAP2, NPC1L1, PLVAP, POMC, PYROXD2 and TRIM29 were significantly associated with *Wald* and *P value*, as shown in Table 3. We further investigated the combined diagnostic efficacy of the seven candidate genes according to the Eq. (9).

$$PP = \frac{1}{1 + e^{-(constant + \sum_1^n coefficient_i * expression_i)}}.$$  (9)

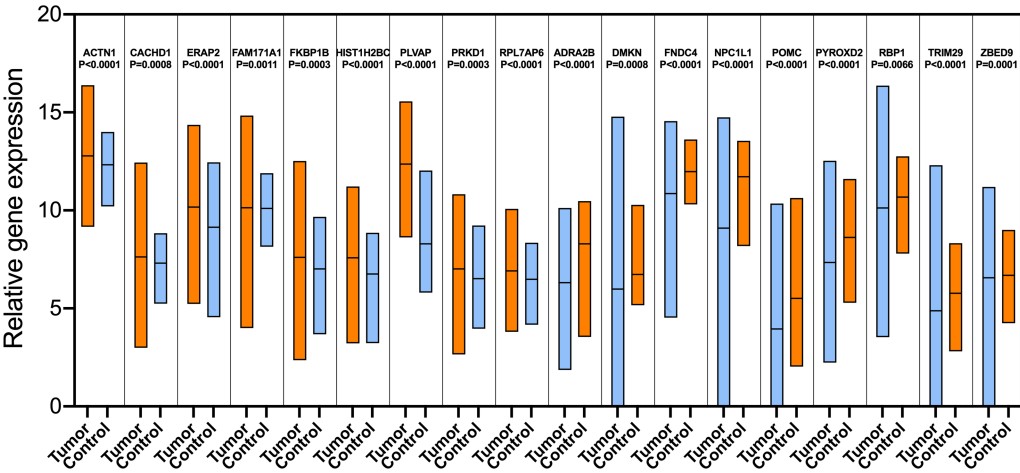

**Figure 2** **The relative expressions of the 18 genes in tumor and control groups.** The values were displayed as floating bars (min to max) with a line at the mean value. The first nine genes increased in tumors (orange) compared to controls (blue), while the last nine genes decreased in tumors (color in blue) compared to controls (orange). The statistic was performed by Wilcoxon rank sum test with continuity correction in R.

**Table 3** **Logistic regression analysis of the independent significance of the 18 genes as diagnostic biomarkers.**

| Variable | Coefficient | Std. Error | Wald | *P* value | Odds ratio | 95% CI |
|----------|-------------|------------|------|-----------|------------|--------|
| **ADRA2B** | −1.2426 | 0.46552 | 7.1249 | 0.0076 | 0.2886 | 0.1159 to 0.7188 |
| **ERAP2** | 0.74916 | 0.33879 | 4.8897 | 0.027 | 2.1152 | 1.0889 to 4.1090 |
| **NPC1L1** | −0.72899 | 0.34876 | 4.3691 | 0.0366 | 0.4824 | 0.2435 to 0.9556 |
| **PLVAP** | 3.42399 | 0.83386 | 16.8608 | <0.0001 | 30.6915 | 5.9872 to 157.331 |
| **POMC** | −0.80808 | 0.32528 | 6.1715 | 0.013 | 0.4457 | 0.2356 to 0.8432 |
| **PYROXD2** | −0.64701 | 0.32983 | 3.848 | 0.0498 | 0.5236 | 0.2743 to 0.9995 |
| **TRIM29** | −0.99483 | 0.49833 | 3.9853 | 0.0459 | 0.3698 | 0.1392 to 0.9821 |
| **Constant** | 1.81243 | 8.70136 | 0.04339 | 0.8350 | | |

In the above Equation, *PP* was the functional formula for predicting the incidence of LIHC, *i.e.,* PP-value, and the *constant* and *coefficient* were the result of logistic regression in Table 3. The results of receiver operating characteristic (ROC) curve analysis using MedCalc software based on the value of *PP* and classification labels of LIHC dataset was shown in the Fig. 4, which had an area under the curve (AUC) greater than 0.999 and $P < 0.0001$. It demonstrated that the seven genes significantly distinguished tumors from normal samples in LIHC dataset.

To explore whether the 18 genes are associated with survival time of phenotype information in LIHC dataset, the Kaplan–Meier (KM) survival curve was performed by using the "survival" and "survminer" packages in R. For each gene, the cut-off points obtaining from "survminer" package then divided gene expression values into the high (high) and the low (low) groups. We identified that higher expression levels of CACHD1,

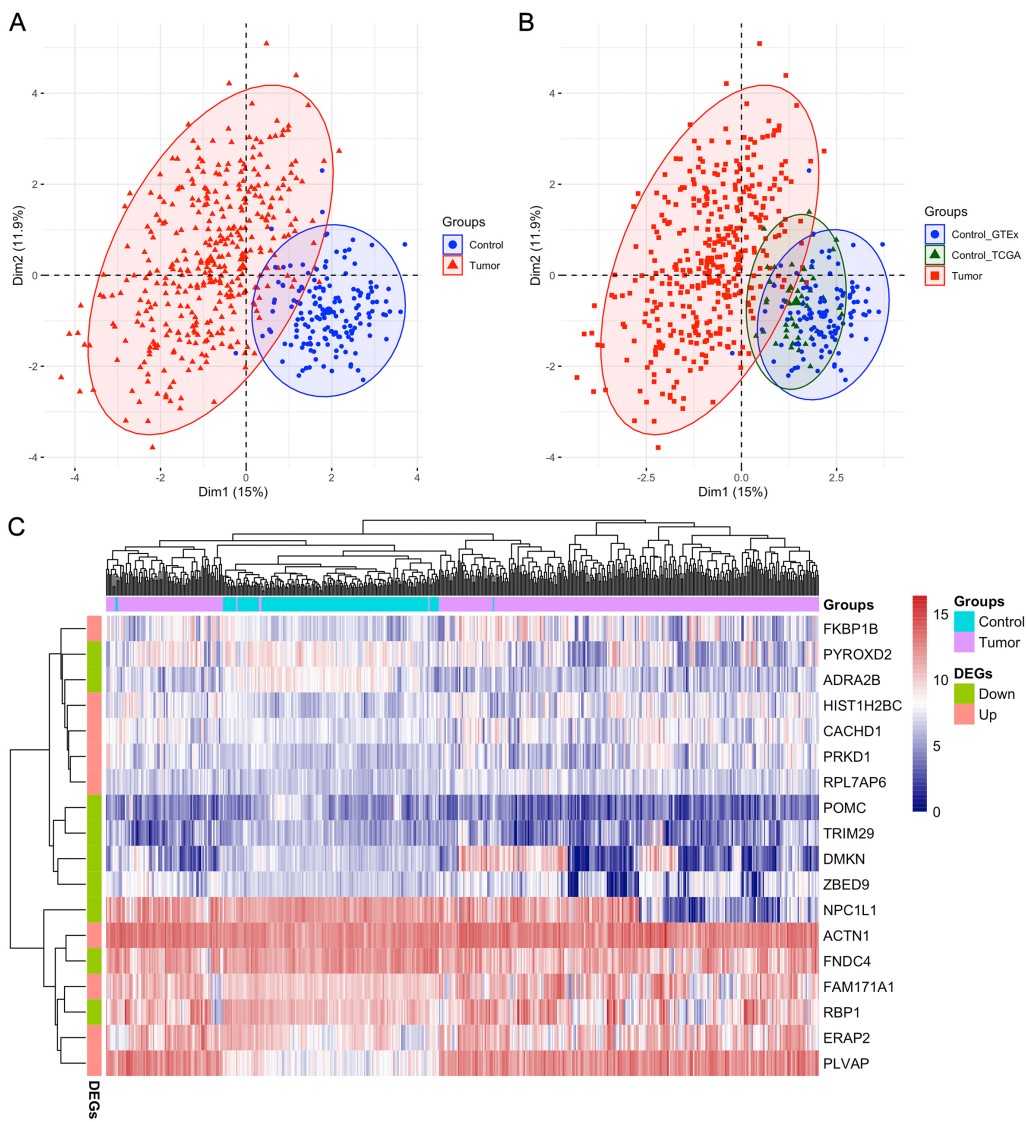

**Figure 3** **PCA and heatmap analysis of the 18 gene signatures obtained from MMPSO method in LIHC dataset.** (A) PCA analysis of tumor and control samples. The PCA analysis was performed by using "FactoMineR" and "factoextra" packages in R; (B) PCA analysis of tumor and control samples, the latter including control_GTEx and control_TCGA samples; (C) Heatmap of all the samples based on the 18 gene expression profiling. The heatmap analysis was performed by using "pheatmap" package in R. DEGs: differentially expressed genes. The expression levels of up DEGs were increased in tumors compared to controls, and the down DEGs were decreased.

FKBP1B, PRKD1, and RPL7AP6 were associated with worse overall survival (OS) time, whereas higher expression levels of ADRA2B, PYROXD2 were associated with better OS, as shown in Fig. 5.

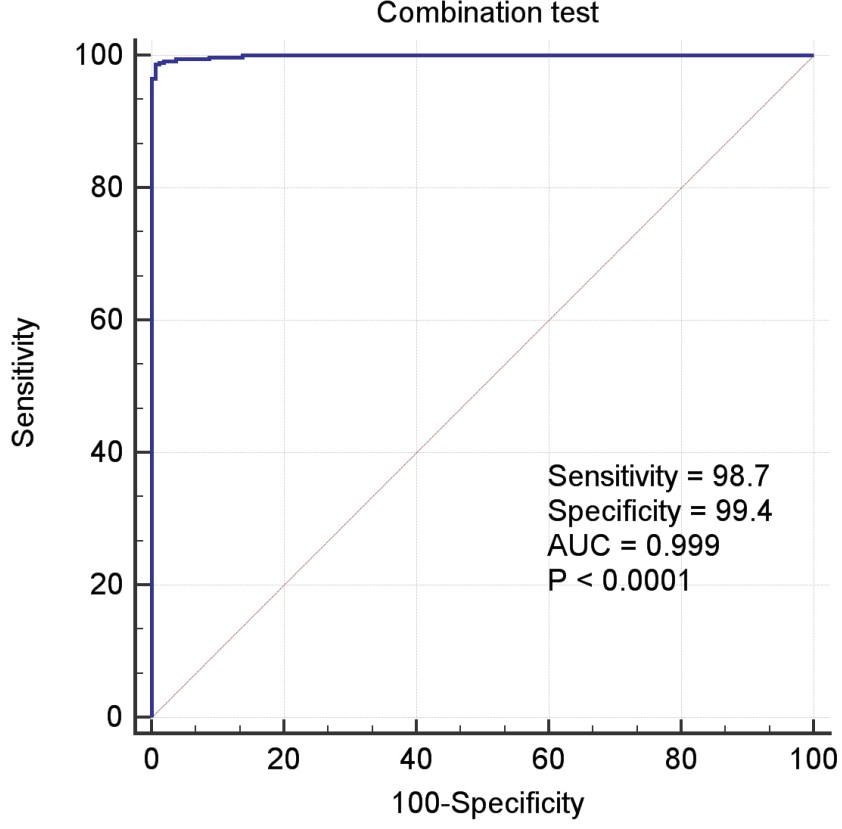

**Figure 4** The ROC curves of the combination of seven candidate biomarkers (ADRA2B, ERAP2, NPC1L1, PLVAP, POMC, PYROXD2 and TRIM29).

## DISCUSSION

High-dimensional data such as text data, multimedia data, aerospace collection data and biometric data have become more common in recent years (*Li, Wu & Li, 2020*; *Saeys, Inza & Larrañaga, 2007*; *Wainwright, 2019*). The need for efficient processing technology for high-dimensional data has become more urgent and challenging. Feature selection, as one of the most popular methods for dimension reduction, plays an important role in high-dimensional data processing, particularly in biological information data (*Nguyen, Xue & Zhang, 2020*; *Xue et al., 2015*).

Generally, features filtered out of the original high-dimensional dataset have more definite physical meanings, making it more convenient for researchers to carry out subsequent work. The direct benefits of feature selection are that it reduces the burden of follow-up work and improves model generalization. Choosing the best subset from the original features has been shown to be a NP-hard problem. When the number of features is $N$, $2^N$ combinations of features must be tried using the greedy strategy, which is unsustainable for ordinary computer systems, especially when the number of features is very large (*Faris et al., 2018*; *Xue et al., 2015*). As a result, over the last few decades, some heuristic algorithms have been proposed to find the best subset that can best represent the

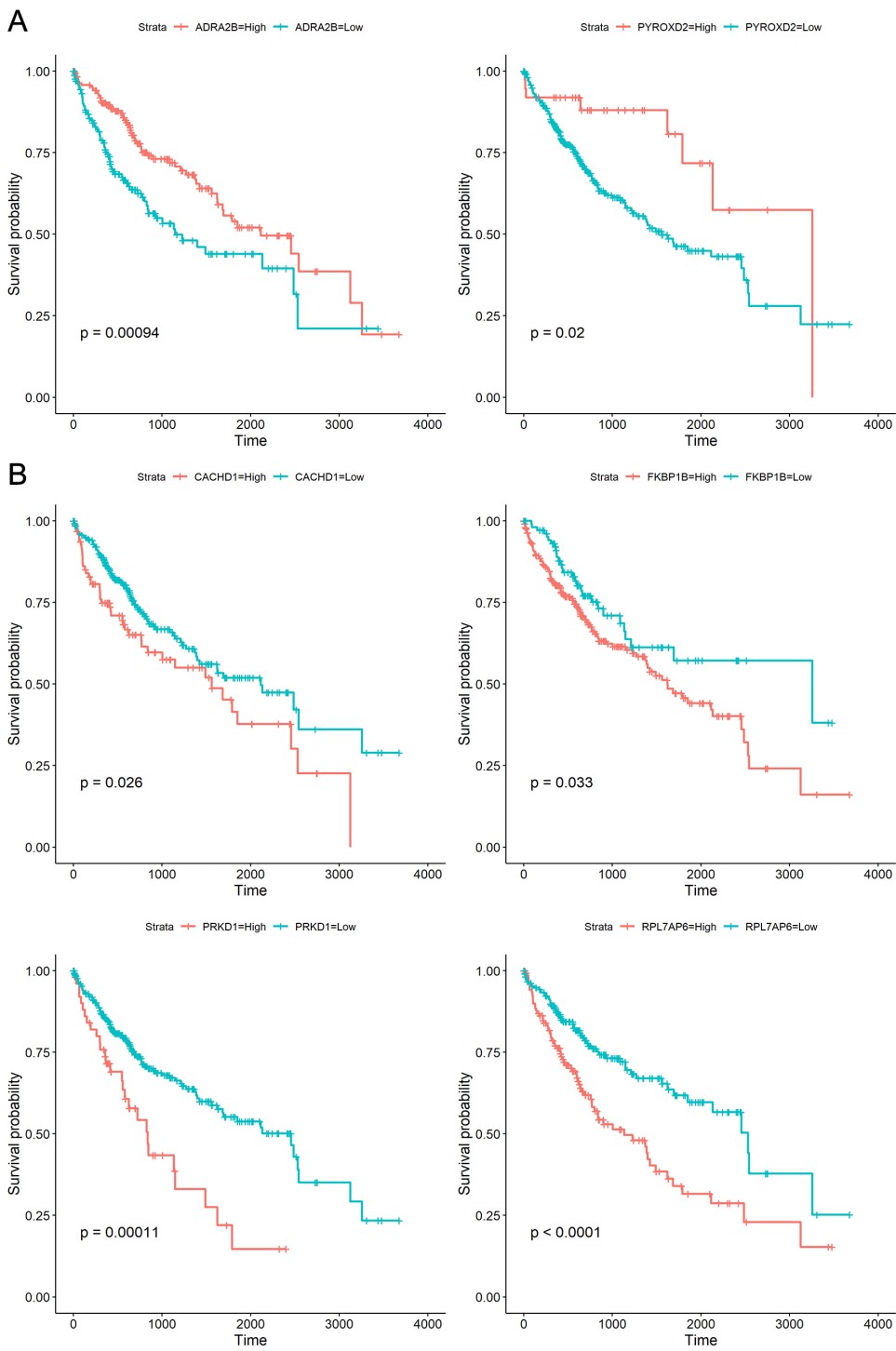

**Figure 5** **Kaplan–Meier curves for prognostic analysis of six genes, including ADRA2B, PYROXD2, CACHD1, FKBP1B, PRKD1 and RPL7AP6.** (A) The Kaplan–Meier curves based on two down-regulated genes (ADRA2B, PYROXD2); (B) the Kaplan–Meier curves based on four up-regulated genes (CACHD1, FKBP1B, PRKD1 and RPL7AP6). The horizontal axis represents the survival time (days), and the vertical axis represents the overall survival rate.

feature meanings of the original dataset. A best subset can be used to represent the original dataset with the least amount of redundancy between features and the highest correlation between the subset's features and labels. The mRMR algorithm, as an implementation of this mind, can obtain the top $K$ ranking features, where the $K$ value must be manually set and mutual information is used to measure the relevance of two features. The mRMR algorithm is undoubtedly an excellent feature selection framework, and it has been widely used in a variety of fields. Despite this, there are still some shortcomings that can be addressed. On the one hand, mutual information can only handle discrete data, which means that continuous data must be discretized in advance, resulting in some accuracy loss. The output of the mRMR, on the other hand, is the top $K$ features, and there is no "golden rule" to specify a suggested or best $K$ value. To address the above two issues, we proposed a hybrid method called MMPSO. First, the noisy data were removed using a conventional method, and the imbalanced data were corrected using random oversampling technology for preprocessing. Second, we used the MIC (*Cao et al., 2021*; *Reshef et al., 2011*) instead of the MI to obtain a more precise correlation value. Although study (*Kinney & Atwal, 2014*) has noted that estimates of mutual information are more equitable than estimates of MIC, there is no denying that MIC has been widely and conveniently used. Furthermore, rapidMic (*Tang et al., 2014*), an algorithm that can use multiple threads simultaneously, was used to reduce the time expenditure of MIC algorithm. Finally, we selected the top $K$ features from the second step as the input for the PSO algorithm to find the best subset. In accordance with the preceding thought, we conducted our experiment using ten datasets. On these datasets, the MMPSO method was applied to compared with the other methods, including mRMR, ILFS, ReliefF, Mutinffs, FSV, Fisher, CFS, UFSOL. We applied LIBSVM library to evaluate the performance of the three methods and test the classification accuracy of the selected features. The experimental results of the ten datasets provided evidences that the MMPSO method performed better than other feature selection algorithms, when all the algorithms used the same number of features. It's worth mentioning that the mRMR performed similarly to the MMPSO method in the previous findings. Here, we still highlighted the advantages of MMPSO, since it can autonomously select feature subset on the basis of MIC, which will improve the accuracy of classification.

To investigate the efficacy of our method for biological data, we used the LIHC dataset (a dataset of RNA expression in liver hepatocellular carcinoma) for further analysis. After performing the MMPSO algorithm, a signature including 18 genes was identified as significant biomarkers for distinguishing between tumor and normal groups. The PCA and ROC analysis results all confirmed that the biomarkers we selected have great discrimination ability, while six biomarkers were significantly associated with the overall survival of patients. Furthermore, we need to discuss the significance of these biomarkers from a biological perspective. Studies (*Wang et al., 2014*) identified and evaluated tumor vascular PLVAP as a therapeutic target for treatment of HCC but not in nontumorous liver tissues, and this result may provide some clues for the development of drugs for patients with HCC. FNDC4 (*Wang et al., 2021*) was reported to be an extracellular factor and played important roles in the invasion and metastasis of HCC in that it promoted the invasion and metastasis of HCC partly *via* the PI3K/Akt signaling pathway.

*Wang et al. (2019)* discovered that PYROXD2 localizes to the mitochondrial inner membrane/matrix, and it plays important roles in regulating mitochondrial function of HCC. TRIM29 plays critical role in many neoplasms. The study (*Xu et al., 2018*) revealed that higher TRIM29 expression was associated with higher differentiation grade of HCC and its depletion promoted liver cancer cell proliferation, clone formation, migration and invasion. The regulatory role of ACTNs in tumorigenesis has been demonstrated and ACTN1 was significantly upregulated in HCC tissue and closely related to tumor size, TNM stage and patient prognoses (*Chen et al., 2021b*). Based on the above studies, the majority of the genes identified by our algorithm are promising candidate biomarkers for the diagnosis or treatment of liver cancer.

## CONCLUSION

In this paper, we proposed the MMPSO hybrid algorithm to identify a feature subset for high-dimensional dataset. The experimental data provided evidences that our method outperformed others. More importantly, by applying our proposed algorithm to the biological LIHC dataset, we obtained the gene signatures in classifying tumors and normal samples with high efficacy. Our study also has several limitations. Despite the fact that we used rapidMic to accelerate the calculation, the computational complexity for too many features remains relatively high. In addition, we selected only the top $K$ features as the PSO input without a theoretical foundation. Therefore, there is still some space for improvement in selecting a better subset of features for high-dimensional datasets, and we will advance this in future works.

### Funding
This work was supported by the National Science Foundation of China (No.61573285). The funders had no role in study design, data collection and analysis, decision to publish, or preparation of the manuscript.

### Grant Disclosures
The following grant information was disclosed by the authors:
The National Science Foundation of China: No.61573285.

### Competing Interests
The authors declare there are no competing interests.

### Author Contributions
- Yangyang Wang performed the experiments, analyzed the data, performed the computation work, prepared figures and/or tables, and approved the final draft.
- Xiaoguang Gao conceived and designed the experiments, authored or reviewed drafts of the paper, and approved the final draft.
- Xinxin Ru performed the experiments, authored or reviewed drafts of the paper, and approved the final draft.

- Pengzhan Sun analyzed the data, prepared figures and/or tables, and approved the final draft.
- Jihan Wang conceived and designed the experiments, prepared figures and/or tables, and approved the final draft.

## Data Availability

The raw data is available in the Supplemental Files.

## Supplemental Information

Supplemental information for this article can be found online at http://dx.doi.org/10.7717/peerj-cs.933#supplemental-information.

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
