# Peer review of "A hybrid feature selection algorithm and its application in bioinformatics"

_PeerJ Computer Science, doi:10.7717/peerj-cs.933_

## Round 0.1 · original submission · Major Revisions

The main concern is from using the same data set. Please follow the reviewers' comments carefully to revise the manuscript carefully. I recommend a major revision with a proof reading before re-submission.

Reviewer 1 ·

Basic reporting

Wang et al. developed a hybrid MMPSO method combining feature ranking and heuristic searching which yield higher classification of accuracy. To demonstrate the accuracy of such method, the authors applied the method to eleven datasets and identified 18 tumor specific genes, nine of which were further confirmed in a HCC TCGA dataset. Together, the study presents a highly effective MMPSO algorithm with great potential to identify novel biomarkers and therapeutic targets for cancer research and treatment.

Experimental design

The study is original and novel in the sense that it develops the hybrid algorithm by combining mRMR and MIC method. The methodology is very concrete as the authors compared the MMPSO method to eight other methods and clearly showed their superiority. Furthermore, the MMPSO was applied to the HCC biological dataset and gave a 18 gene signature which successfully separate 531 samples into cancer and normal groups. Many of the targets in the signature were validated by published research and/or their own Kaplan Meier method. Therefore, the study is well targeted at the Aims of Scope of Peer J Computer Science.

Validity of the findings

The findings in this study are thoroughly validated. The conclusions are well stated.

Reviewer 2 ·

Basic reporting

The authors aimed to develop an improved feature selection algorithm. They applied the algorithm to the public gene expression dataset and found some candidate biomarker genes for classifying tumor versus normal samples. This is an incremental work based on previous methods and has potentially interesting applications on tumor diagnosis. However, the following concerns should be addressed to meet the basic requirement of a bioinformatics method paper.

Experimental design

Major concern #1: the authors used the same dataset (LIHC) for selecting the tumor/normal gene signatures and testing their performance. This does not prove the superiority of the algorithm since it can simply overfit this dataset. I suggest 1) selecting features based on a subset of samples and validating on the left-out samples; 2) using another cohort of normal/tumor gene expression datasets for testing the selected features.

Major concern #2: the authors did not compare the gene signatures selected by their method versus other methods. For instance, using the top differential genes of a simple differential gene expression analysis with DESeq2 or EdgeR, do they separate the tumor samples better than these 18 genes? Given that in many of the ten datasets in Figure 1 and Table 2, mRMR performs very similarly to MMPSO, what is the performance on the LIHC dataset if only using mRMR to select features?

Validity of the findings

Major concern #3: In line 210, the authors listed the 18 gene signatures and “nine genes were significantly upregulated in tumors compared to normal samples” but did not clarify which genes were differentially upregulated or downregulated and what were the criteria. The authors also did not provide method detail of how they determined the high and low expression levels for sample stratification in Figure 4. For instance, I checked the average values of some of the genes in the two raw data files (data_tumor.csv, data_normal.csv) provided by the authors:

ACTN1, ENSG00000072110 Tumor = 12.78586 Normal = 12.33047
CACHD1, ENSG00000158966 Tumor = 7.624089 Normal = 7.308306

In lines 292-296, the authors discussed the literature that a higher level of ACTN1 is associated with HCC size, and in Figure 4, CACHD1 was described as an ungulated gene associated with worse prognosis, however, I do not consider a difference of < 0.5 as significantly different in gene expression. At the very least, the authors should provide a box-whisker plot with proper statistics of the expression of these 18 genes across the normal/tumor samples they used to demonstrate their points.

Additional comments

Minor comments. Please read carefully for typos. Line 18: “a optimal subset”. Line 70: “Yuanyuan et al. 2021”. In line 17, MMPSO should not be abbreviated because this is its first appearance in the whole manuscript.

---

## Round 0.2 · accepted · Accept

All the comments have been addressed well in the revised version. Therefore, I recommend accepting this manuscript based on the solid contribution.

Reviewer 2 ·

Basic reporting

The authors have properly addressed the questions I raised.

Experimental design

The authors have properly addressed the questions I raised.

Validity of the findings

The authors have properly addressed the questions I raised.